Original research

# How do high ambient temperatures affect infant feeding practices? A prospective cohort study of postpartum women in Bobo-Dioulasso, Burkina Faso

Chérie Part ![ORCID],[1] Véronique Filippi,[2] Jenny A Cresswell,[2] Rasmané Ganaba,[3] Shakoor Hajat,[1] Britt Nakstad,[4,5] Nathalie Roos ![ORCID],[6] Kadidiatou Kadio,[7] Matthew Chersich,[8] Adelaide Lusambili ![ORCID],[9] Seni Kouanda,[7] Sari Kovats[1]

**Correspondence to**
Dr Chérie Part;
cherie.part@lshtm.ac.uk

## ABSTRACT

**Objective** To examine the effects of high ambient temperature on infant feeding practices and childcare.

**Design** Secondary analysis of quantitative data from a prospective cohort study.

**Setting** Community-based interviews in the commune of Bobo-Dioulasso, Burkina Faso. Exclusive breastfeeding is not widely practised in Burkina Faso.

**Participants** 866 women (1:1 urban:rural) were interviewed over 12 months. Participants were interviewed at three time points: cohort entry (when between 20 weeks' gestation and 22 weeks' postpartum), three and nine months thereafter. Retention at nine-month follow-up was 90%. Our secondary analysis focused on postpartum women (*n*=857).

**Exposure** Daily mean temperature (°C) measured at one weather station in Bobo-Dioulasso. Meteorological data were obtained from publicly available archives (TuTiempo.net).

**Primary outcome measures** Self-reported time spent breastfeeding (minutes/day), exclusive breastfeeding of infants under 6 months (no fluids other than breast milk provided in past 24 hours), supplementary feeding of infants aged 6–12 months (any fluid other than breast milk provided in past 24 hours), time spent caring for children (minutes/day).

**Results** The population experienced year-round high temperatures (daily mean temperature range=22.6°C–33.7°C). Breastfeeding decreased by 2.3 minutes/day (95% CI -4.6 to 0.04, *p*=0.05), and childcare increased by 0.6 minutes/day (0.06 to 1.2, *p*=0.03), per 1°C increase in same-day mean temperature. Temperature interacted with infant age to affect breastfeeding duration (*p*=0.02), with a stronger (negative) association between temperature and breastfeeding as infants aged (0–57 weeks). Odds of exclusive breastfeeding very young infants (0–3 months) tended to decrease as temperature increased (OR=0.88, 0.75 to 1.02, *p*=0.09). There was no association between temperature and exclusive breastfeeding at 3–6 months or supplementary feeding (6–12 months).

**Conclusions** Women spent considerably less time breastfeeding (~25 minutes/day) during the hottest, compared with coolest, times of the year. Climate change adaptation plans for health should include advice to breastfeeding mothers during periods of high temperature.

## STRENGTHS AND LIMITATIONS OF THIS STUDY

⇒ This is the first study to quantify acute effects of ambient heat on breastfeeding behaviour.

⇒ Multi-stage stratified sampling was used to select a population-representative cohort of pregnant and postpartum women in the commune of Bobo-Dioulasso, Burkina Faso.

⇒ Detailed questionnaires enabled extensive confounder control.

⇒ Outcome measures relied on self-reports, including time-use estimations. However, questions were embedded within an extensive interview schedule, reducing the likelihood of response bias, and measures were used to assist participants with time estimations.

⇒ The small sample size and short recruitment window may have limited our ability to detect statistically significant associations.

## INTRODUCTION

Climate change is a growing threat to population health in Africa,[1 2] with heatwaves increasing in severity and duration, especially in the Sahel.[3] Maternal and neonatal health will be affected through the adverse effects of heat on preterm birth,[4 5] stillbirth[4 5] and maternal nutrition.[6] Child wasting and malnutrition are expected to increase.[2] High temperatures may also reduce cognitive function[7] and interfere with daily activities, leading to a decline in emotional health and well-being.[8] Mothers may find it difficult to breastfeed their infants under extreme heat[9] and may also change their behaviour due to perceived risks to health. For example, there is still a common misconception among postpartum women in several African countries that breast milk is not sufficient to hydrate babies during hot weather, leading to supplementary feeding of infants (with sometimes

non-potable water)[10–13] and a reduction in exclusive breastfeeding.[10]

Breastfeeding and, in particular, exclusive breastfeeding has well-established benefits for child health and development.[14–16] Breastfeeding reduces the risk of diarrhoea and respiratory infections among infants, and is associated with a higher IQ and reduced obesity in later life.[16] There are also benefits for maternal health, with nursing mothers at lower risk of breast and, potentially, ovarian cancers.[16] The WHO recommends that infants be fed with breast milk exclusively for the first 6 months and that no solids or other liquids are given during this period, including water.[17] However, the self-reported prevalence of exclusive breastfeeding is low in many African countries.[16 18] In Burkina Faso, less than 25% of women reported exclusive breastfeeding of their young infants (less than 6 months old) in 2010–2015.[18]

It is not unusual for breastfeeding patterns to change in hot weather. Infants may refuse to feed during the hottest part of the day, or they may demand more frequent, but shorter, feeds throughout the day.[19] In doing so, babies consume mostly low-fat milk (foremilk) and avoid breast milk with a high fat content (ie, afternoon/evening milk and hindmilk).[19 20] Mothers must change their breastfeeding patterns to accommodate their infants' needs and may spend more time breastfeeding as temperatures rise. Conversely, women may spend less time breastfeeding during periods of high temperature due to increased discomfort for both mother and child,[9] increased provision of water (believed necessary to quench baby's thirst in some African settings)[21] and/or associated health effects, such as low energy[8] and heat exhaustion.[22]

Infants and young children are particularly vulnerable to heat injury and dehydration due to a greater surface area to body mass ratio.[23] Therefore, as temperatures rise in hot climates, mothers may spend more time watching over their children and other children in the household, keeping them hydrated and tending to them when unwell. Such increased demands on time may cause difficulties for mothers in low-income countries such as Burkina Faso, where women work to supplement household income (particularly in agriculture, horticulture and small trade), as well as undertaking important domestic responsibilities (including gathering food, water, fuel and feeding livestock).[6] Most women work in the informal sector[24]; therefore, paid maternity leave is uncommon, and many women return to work early in the postpartum period.

Average monthly temperatures in Burkina Faso range between 25°C and 33°C,[25] and the impacts on infant care practices are largely unknown. Studies in South America, South Asia and Africa show seasonal differences in breastfeeding behaviour, with conflicting results.[26–29] For example, in Bihar, India, infants under 6 months were more likely to be exclusively breastfed in the cooler than warmer season,[29] whereas, in rural Egypt, exclusive breastfeeding of infants aged 6–11 months was more prevalent in the hot than cool season.[26] However, such studies are not sufficient to demonstrate an effect of ambient temperature as the competing time demands of women's domestic and agricultural workloads,[10 30–33] as well as other potentially important drivers (eg, household food security), also vary with season and weather in rural settings.[33 34] With daily temperatures in West Africa expected to exceed 50°C in some regions,[35] further research is essential so that maternal and child health programmes can be updated.

This study aims to explore the effects of daily outdoor temperature on infant feeding practices and childcare in western Burkina Faso. We hypothesised that (1) time spent breastfeeding is associated with same-day temperature; (2) women are less likely to breastfeed exclusively as temperatures rise; (3) women are more likely to provide supplementary fluids as temperatures rise; and (4) time spent caring for children increases with same-day temperature.

## METHODS

We undertook secondary analyses of quantitative data from an observational prospective cohort study of pregnant and postpartum women in Bobo-Dioulasso, Burkina Faso (the PopDev study), which aimed to assess the impacts of pregnancy on income-generating and non-income-generating activities among women in Burkina Faso, and to identify interventions to increase household income.[36]

### Participants

Multi-stage stratified (urban vs rural) sampling was used to select a population-representative cohort of pregnant and postpartum women in the commune of Bobo-Dioulasso. The 2006 census was used as the sampling frame to select locality clusters. It was estimated that each cluster must contain a minimum of 300–330 households to identify 30 eligible participants per cluster. Thirty-eight locality clusters were identified (14 urban, 24 rural), and participants were recruited within households at cluster level. Households were visited per selected cluster using a modification of the WHO's Expanded Programme on Immunisation sampling methodology.

Women were eligible to participate in the PopDev study if aged 15–45 years and between 7 months' gestation and 3 months' postpartum at recruitment.[36] Sixty-two women did not meet the criteria for PopDev but were retained in the dataset for secondary analyses. The full dataset comprised 866 women aged 14–47 years who were between 20 weeks' gestation and 22 weeks' postpartum at baseline. All women who completed at least one interview postpartum (*n*=857) were included in the secondary analysis (see figure 1).

### Setting

The commune of Bobo-Dioulasso is predominantly urban, and includes the second largest city in Burkina Faso (Bobo-Dioulasso) with approximately 900 000

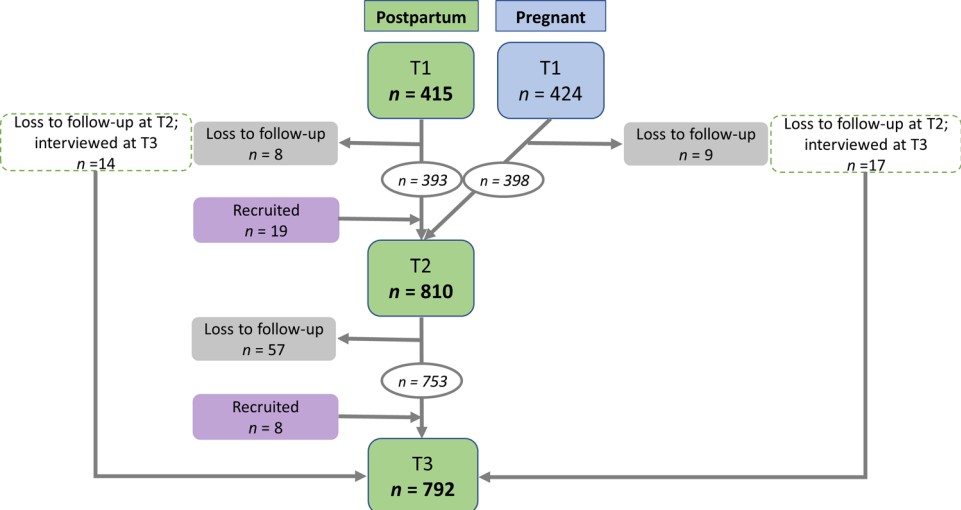

**Figure 1** Sampling flowchart for secondary analysis showing the number of interviews conducted with pregnant (blue) and postpartum women (green) at each interview round (T). Data from all interviews highlighted in green were included in the secondary analyses.

inhabitants.[37] Small settlements and villages, with mainly an agricultural focus, are located in rural areas surrounding the large urban centre. All rural participants in this study resided within 40–50 km of the city.

The commune has a tropical savannah climate,[38] with two distinct seasons: dry (November–May) and rainy (June–October). During the dry season, average temperatures are highest in March–May and lower in November–February.

### Data collection

Participants were interviewed in their homes at three time points: cohort entry, and three and nine months thereafter. Retention at the nine-month visit was 90%. Several attempts were made to interview each participant at each interview round. The reason for non-interview was recorded, when possible. Eight hundred and 39 participants were recruited immediately before/during the first round of interviews (29 November 2013–23 March 2014). Twenty-seven women were recruited during the second or third round of interviews (4 March–9 September 2014 and 2 September–12 November 2014, respectively). All interviews occurred between 29 November 2013 and 12 November 2014, using structured questionnaires to ensure the same wording of questions for all participants. Interviews were conducted in local languages (predominantly Dioula) and homogeneity in translations was verified during interviewer training. All questions were designed to be non-leading.

During each interview, the participants were asked to recall their activities on the previous day (or two days previous when the day before the interview was atypical), and how many minutes they had spent on each activity. The recall period was defined as 'between waking up yesterday morning and waking up this morning'. Participants described their activities and the interviewer categorised them using a predefined list, which was

added to when needed. Breastfeeding, caring for children (bathing/dressing, feeding, playing/watching, and tending to when unwell), income-generating work (eg, agroprocessing for trade, sale of products at market, small business, and office activities), attending classes (eg, literacy courses) and household chores (eg, preparing meals, cleaning clothes, washing dishes, fetching water, and fetching fuel) were included in this list.

To assist participants in time-use recall, women were initially asked to list all activities that they had engaged in during a specific time of day (eg, between waking up and midday). Participants were then asked probing questions to assist them in estimating the duration of each activity. For example, some women said that they woke with the call of the muezzin, which enabled the interviewer to determine time of wake. Participants would then be asked if they began their first listed activity immediately or if they did something else first. The interviewer asked when their first activity ended to which women responded, 'to the first rays of sunshine', for example. Thus, the interviewer had adequate information to estimate time duration of the first activity. Once participants had finalised their time-use estimates for the specified period, the process was repeated for the next time of day, until the full 24-hour recall period was complete. Participants used a notebook to draft and revise time-use estimates before final responses were recorded in the questionnaire. Indications of time (particularly the path of the sun), combined with current events of life, served as benchmarks for time estimations.

At first interview following childbirth (interview round 1 or 2), participants were asked how many children were born and the date of delivery. At interview rounds 2 and 3, women were asked if they were still breastfeeding their baby, if their baby had anything else to drink in the past 24 hours and which (if any) fluids had their baby been

given to drink in the past 24 hours. These questions were used to construct binary study outcomes of exclusive breastfeeding (still breastfeeding and no fluids other than breast milk provided in the past 24 hours) and supplementary feeding (any fluids other than breast milk provided in the past 24 hours).

Interviews included additional questions to those described herein in order to fulfil the aims of the PopDev study. Each interview lasted approximately 45–60 minutes. Questionnaires are available online (https ://datacompass.lshtm.ac.uk/id/eprint/64/).

### Meteorological data

Daily meteorological data (mean, minimum and maximum temperature (°C), relative humidity (%) and wind speed (km/hour)) were obtained from TuTiempo. net[39] for a single weather station in Bobo-Dioulasso, located in the industrial district (Zone Industrielle) (11°09′36.0″N 4°18′36.0″W). Eleven days of temperature data were missing over the study period and were excluded from the analysis.

### Outcomes

Four outcomes were assessed: (1) breastfeeding duration: self-reported time spent breastfeeding on the day/night before interview (total minutes in 24 hours); (2) exclusive breastfeeding: no liquids other than breast milk given in the past 24 hours; (3) supplementary feeding: any liquid other than breast milk given in the past 24 hours; and (4) childcare duration: self-reported time spent exclusively on childcare (including bathing/dressing, feeding, playing/watching, and tending to when unwell) on the day/night before the interview (total minutes in 24 hours). Breastfeeding duration, exclusive breastfeeding and supplementary feeding outcomes referred specifically to the target (newborn) infant. Childcare duration did not refer specifically to the newborn infant.

### Exposure

The primary exposure was daily mean temperature (°C). Daily mean temperature correlated strongly with daily minimum ($r=0.8$, $p<0.001$) and maximum temperatures ($r=0.89$, $p<0.001$), and was considered the best approximation of overall exposure during the recall period. Apparent daily mean temperature (°C) was calculated from daily mean temperature, relative humidity and wind speed, using the R HeatStress package,[40] to test the robustness of our findings.

### Data analyses

Daily exposures were linked with outcomes at individual level, by date of interview minus 1 day ($t$–1) to reflect same-day temperature when activities were undertaken. Categorical outcomes, potential confounding variables and covariates (see online supplemental table S1) were summarised as proportions (expressed as percentages). Continuous variables were summarised as mean±SD if normally distributed, or as median and IQR if not normally distributed. Summary statistics were stratified by interview round where applicable.

The functional form of temperature–time use (breastfeeding and childcare) associations were determined by aggregating outcome data to daily level, fitting natural cubic splines of time (to adjust for seasonal patterns and trends unrelated to temperature), and examining locally weighted smoothing of Pearson's standardised residuals from the fitted splines plotted against daily mean temperature.

Multilevel linear regression was then used to estimate the effects of daily mean temperature on time spent (1) breastfeeding and (2) caring for children. This approach made use of all available time-use data on breastfeeding and childcare, while accounting for the longitudinal and nested structure of these data. Interview contacts (level 1) were nested within individual participants (level 2), nested within the locality clusters from which the population was sampled (level 3). Each level was defined as a random coefficient with random intercept to allow for correlation within individuals and clusters. A first-order autoregressive correlation structure allowed for unequal spacing of interviews.

Separate models were developed for each outcome and were adjusted for interview round. Indicator terms were included for calendar month of interview to adjust for season and long-term trends. Adjusting for month (rather than season) of interview provided tighter control of possible confounders, such as household food security and fasting during Ramadan. Other covariates (number born (singleton or multiple birth), infant age (weeks), maternal age (≤19, 20–34, or ≥35 years), gravidity (1, 2–5, or ≥6 pregnancies), residential area (urban or rural), living arrangements (with partner full-time, with partner periodically, not with partner, or not in a relationship), paid work or education (minutes/day), domestic work (minutes/day) and roofing materials (natural, rudimentary, or contemporary); see online supplemental table S1) were added to the models one-by-one, following a forward stepwise process, and were retained in the model if they were significantly associated with the outcome ($p<0.05$), improved model fit (reduced the Akaike information criterion by ≥2%) and/or changed the temperature effect by ≥10%. Cases with missing data were excluded from the analysis. Participants lost to follow-up were included in the analysis.

We restricted our analyses of exclusive breastfeeding and supplementary feeding to infants aged less than 6 months and 6–12 months, respectively. This follows from WHO's recommendations that infants are breastfed exclusively for the first 6 months of life, and that supplementary foods are only introduced thereafter.[17] Data were available for two time points (interview rounds 2 and 3). To reduce model complexity, these outcomes were analysed cross-sectionally, at single time points: exclusive breastfeeding at interview round 2 and supplementary feeding at interview round 3, based on the age range of infants at each round.

Logistic regression was used to test for associations between mean temperature and the odds of (1) exclusive breastfeeding and (2) supplementary feeding, adjusting for month of interview, and other important confounders and covariates following the same stepwise process described earlier. As the effects of temperature on exclusive breastfeeding may change as infants age,[29] our exclusive breastfeeding analysis was age-stratified (<3 months and 3 to <6 months). Interactions between mean temperature and (1) infant age, (2) urban/rural residence and (3) roofing materials of the home were tested in all models (both multilevel and logistic).

Sensitivity analyses involved respecifying models with (1) alternative levels of seasonal control (indicator variables for 'season' rather than 'month'; natural cubic splines of calendar time with 3 knots); and (2) apparent, rather than observed, daily mean temperature.

Analyses were done in R 4.0.4,[41] using RStudio and the following R packages: lme4,[42] nlme,[43] stats,[41] splines,[41] effects,[44] ggplot2,[45] HeatStress[40] and Hmisc.[46]

## Patient and public involvement

Stakeholders were involved during development of the original proposal in June 2012. The objectives and plans for the primary (PopDev) study were discussed with representatives from the community and reproductive health non-governmental organisations (NGOs), as well as health professionals and policy makers at local and national levels, by means of a workshop, email and telephone.

Stakeholders from the policy or associative arena presented their policies at the workshop to inform a group discussion on how best the study objectives could respond to their information needs. In another exercise, stakeholders were asked to identify one positive, one negative and one surprising thing about the proposed study, which yielded particularly useful information when developing the proposal. We received feedback on substantive and methodological aspects of the project, and on communication issues, which was used to shape the study objectives and methodologies.

The interview schedule was piloted with members of the community. Feedback on interview duration, meaningfulness and clarity of questions, and perceived gaps was used to refine the wording of questions and to add/remove items. At the end of the primary (PopDev) study, a stakeholder consultation workshop took place to discuss the findings and their implications for cross-sectoral interventions involving policy makers from different ministries and NGO staff.

Pregnant and postpartum women, as well as community members, in the Kaya and Bogodogo health districts of Burkina Faso were involved before the secondary study began. In-depth interviews with pregnant and postpartum women (n=40) and focus group discussions with community members were undertaken in October–November 2020.[9] The objectives for the secondary analysis were developed and informed by the lived experiences of postpartum women reported during this qualitative work. Specifically, women described how hot weather impedes breastfeeding due to excessive sweating and the discomfort of both mothers and their babies.[9]

Qualitative findings were discussed with stakeholders in maternal and neonatal health, climate change adaptation, as well as pregnant and postpartum women and community members, during a codesign workshop in Ouagadougou, Burkina Faso. Here, breastfeeding messaging was highlighted as an important area of focus for future research and for interventions aimed at reducing the impact of high temperatures on childbearing women and their newborn infants. We will continue our engagement with community members in the Kaya and Bogodogo health districts and will disseminate our findings through meetings, written summaries and audiovisual materials. We will also engage with health decision-makers and provide summaries of the evidence and targeted policy briefs to support decision-makers in actions to reduce the impact of high temperatures on maternal and neonatal health.

## RESULTS

The population experienced year-round high temperatures, with an intra-annual range in daily mean temperature of 22.6°C–33.7°C. Online supplemental figure S1 shows daily minimum and maximum temperatures throughout the study period.

Eight-hundred and fifty-seven participants birthed 881 children (833 singleton births and 24 twin births). Six stillbirths, eight deaths in live born children, and 18 deaths in infants of unknown status at birth were reported. The mean age of women at recruitment was 26.9 years (SD=6.2 years), with a median gravidity of 3 pregnancies (IQR=2–5 pregnancies). Only 33 women (3.9%) were formally employed at baseline, of which 21 women were eligible for (or benefiting from) maternity leave. Informal paid work was more common (see table 1). Most women (808 of 839 interviewed at baseline) lived in houses with contemporary roof materials, primarily sheet metal (731 women) and timber (81 women).

Median time between first and second interviews was 92 days (IQR 80–108 days), and 149 days (IQR 141–155 days) between the second and third interviews. Total median follow-up time was 236 days (IQR 227–257 days). One woman refused to participate during the second round of interviews and one woman was travelling during the third round. The reasons why other women were lost to follow-up (n=74) are unknown.

The vast majority of postpartum women reported breastfeeding their infants at each interview round (table 1). However, the incidence of exclusive breastfeeding was low. Only 148 of 710 infants (20.8%) aged less than 6 months were exclusively breastfed on the day/night before interview 2, and only 11 of 157 (7%) were exclusively breastfed before interview 3.

**Table 1** Cohort characteristics, activities and average daily temperature at each interview round

| | Interview round | | |
| --- | --- | --- | --- |
| | **1** | **2** | **3** |
| Total interviewed (N) | 839 | 810 | 792 |
| **Cohort characteristics** | | | |
| % urban (N) [NA]* | 49.9 (419) [0] | 49.3 (390) [19] | 49.7 (389) [10] |
| % postpartum (N) | 49.5 (415) | 100 (810) | 100 (792) |
| % working in informal sector (N) [NA]* | 37.6 (315) [1] | 42.5 (343) [3] | 48.9 (386) [2] |
| **Living arrangements** | | | |
| % with partner full-time (N) | 80.9 (679) | 80.1 (649) | 80.1 (634) |
| % with partner periodically (N) | 6.9 (58) | 6.4 (52) | 5.3 (42) |
| % not living with partner (N) | 2.5 (21) | 4.3 (35) | 3.2 (25) |
| % not in a relationship (N) | 0 (0) | 8.8 (71) | 7.7 (61) |
| % unknown (N) | 9.7(81) | 0.4 (3) | 3.8 (30) |
| **Postpartum women only** | | | |
| % breastfeeding (N) [NA]* | 100 (408) [7] | 99.7 (782) [26] | 99.7 (765) [25] |
| % supplementary feeding (N) [NA]* | | 80.2 (628) [27] | 98.0 (752) [25] |
| **Time use (self-reported minutes/day)** | | | |
| Breastfeeding, median (IQR) [NA]* | 120 (80–180) [11] | 180 (120–180) [23] | 240 (121–240) [26] |
| Childcare, median (IQR) [NA]* | 30 (15–40) [5] | 30 (20–40) [11] | 20 (15–30) [21] |
| Paid work/education, median (IQR) | 0 (0–0) | 0 (0–92) | 300 (0–420) |
| Domestic work, median (IQR) | 180 (110–240) | 215 (145–300) | 180 (130–235) |
| Infant age (weeks), median (IQR) [NA]* | 5.7 (2.6–9.9) [17] | 12.6 (6.4–19.0) [25] | 33.9 (27.1–41.0) [31] |
| Daily mean temperature (°C), median (range) [NA]* | 27.9 (22.7–32.8) [0] | 27.0 (22.9–33.7) [11] | 27.2 (23.3–30.3) [0] |

Number (N) of women and % of total interviewed at each survey round, or summary statistics specified. Where NA is not provided, N=0.
*Missing values were excluded from calculations.
NA, N missing.

On average, daily breastfeeding duration increased over time (figure 2A). After adjusting for long-term trends, a slight decrease in breastfeeding duration was observed as temperatures increased (figure 3A). Before adjusting for potential confounders (accounting only for the longitudinal and nested structure of the data), breastfeeding was estimated to decrease by 5.6 minutes/day (95% CI -7.0 to -4.1, $p<0.001$, n=783 women) per 1°C increase in same-day mean temperature. After controlling for important confounders (interview round, month of interview, singleton/multiple birth, residential area and minutes/day spent on paid work or education), breastfeeding was estimated to decrease by 2.3 minutes/day (-4.6 to 0.04, $p=0.05$, n=783 women) per 1°C increase in same-day mean temperature (online supplemental table S2). This estimate was for infants aged 0.6–57 weeks

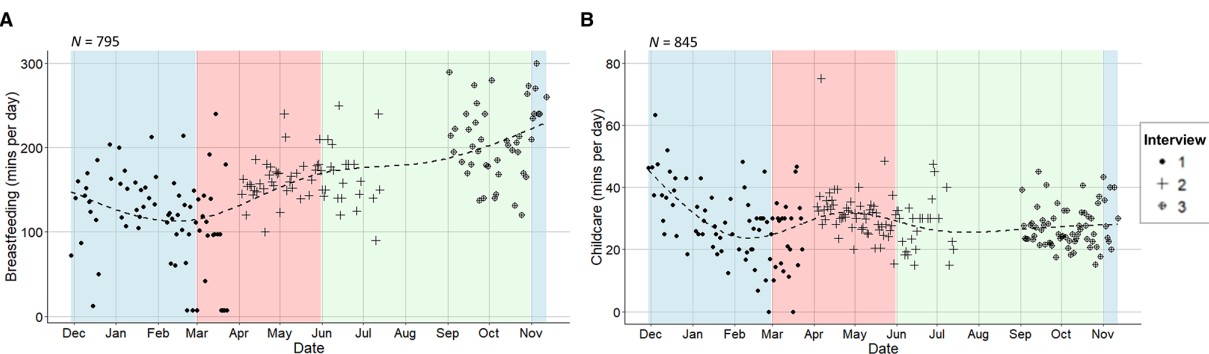

**Figure 2** Average time spent (self-reported minutes/day) (A) breastfeeding and (B) caring for children over time, with fitted natural cubic splines of time (dashed lines). Blue shading indicates the dry cooler season (November–February); red shading the dry hot season (March–May); and green shading indicates the rainy season (June–October). N denotes the sample size. Data source: PopDev study.[36]

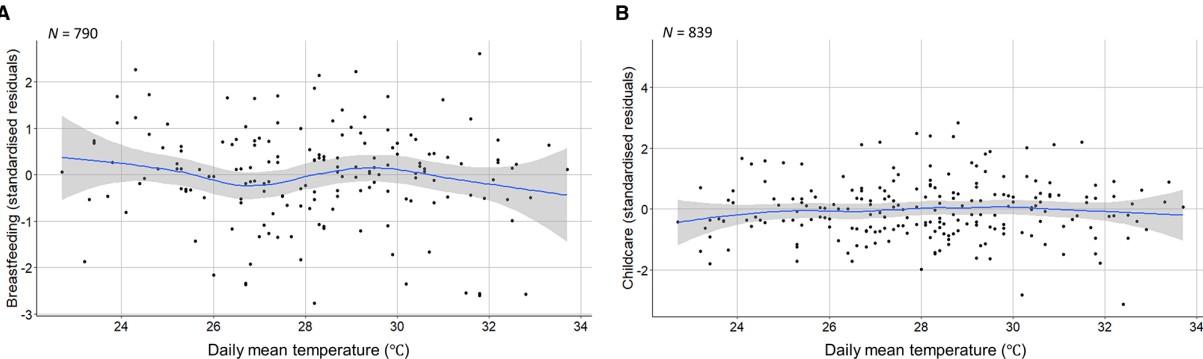

**Figure 3** Scatter plots of daily mean temperature (°C) and standardised residuals from fitted trends (natural cubic splines of time) in the (A) breastfeeding and (B) childcare time series, with locally weighted smoothing (blue line) and 95% CIs (grey shading). *N* denotes sample size.

(median=18.6 weeks). However, temperature interacted with infant age to affect breastfeeding duration (*p*=0.02). Time spent breastfeeding very young infants (4 weeks) did not change with temperature. As infants aged, women were predicted to spend increasingly less time breastfeeding at high temperatures (figure 4).

On average, women spent less time on childcare at interview 3 (table 1), which coincided with the rainy and early dry/cooler seasons (figure 2B). After seasonal control, a slight increase in childcare time was observed as temperatures increased (figure 3B). Before adjustment, we estimated a 0.4 minute increase (95% CI 0.1 to 0.8, *p*=0.02, *n*=814 women) in daily childcare per 1°C increase in mean temperature. We estimated a 0.6 minute increase (0.06 to 1.2, *p*=0.03, *n*=787 women) in childcare per 1°C increase in temperature after adjusting for interview round, calendar month, singleton/multiple birth, infant age, maternal age, women's living arrangements, and time spent on paid work or education (minutes/day) (online supplemental table S3).

There was suggestive evidence that very young infants (<3 months) were less likely to be exclusively breastfed as temperatures increased (unadjusted OR=0.88, 95% CI 0.76 to 1.02, *p*=0.08, *n*=338; adjusted OR=0.88, 0.75 to 1.02, *p*=0.09, *n*=331). However, there was no evidence that daily mean temperature affected the odds of exclusive

breastfeeding at 3–6 months, either before (0.98, 0.80 to 1.20, *p*=0.8, *n*=237) or after adjustment (1.13, 0.86 to 1.52, *p*=0.4, *n*=235). Variability in daily mean temperature was similar for both groups (<3 months=24.1°C–33.7°C, 3–6 months=25.3°C–33.7°C); however, the rate of exclusive breast feeding was higher among women with younger (<3 months=30% (120 of 396 women)) than older infants (3–6 months=9% (28 of 312 women)).

A large proportion of women provided supplementary fluids to their infant (table 1): primarily water, herbal tea and, in the rainy season, boiled water. Milk other than breast milk was rarely given. By 6–12 months, the provision of supplementary fluids was almost universal (99.2% (605 of 610) infants). Therefore, analysis of association with daily mean temperature was not feasible.

There was no evidence of an interaction between temperature and residential area (urban/rural) or type of roofing materials on any outcome measured (*p*>0.05).

Estimated temperature effects were generally robust to sensitivity analyses. The main effect on breastfeeding duration was very robust and increased in statistical significance with alternative methods of seasonal control. The interaction effect between temperature and infant age on breastfeeding duration and the main effect of temperature on daily childcare duration were also fairly robust, but with reduced significance. Redefining the exposure as apparent ('feels like') daily mean temperature did not change the estimated effect on exclusive breastfeeding of very young infants (<3 months), but increased the statistical significance of this finding.

## DISCUSSION

This study explored the impacts of high ambient temperature on infant feeding practices among postpartum women in a low-income setting. We found a decrease in breastfeeding duration as temperatures increased, approximating to a 25 minute reduction in breastfeeding on the hottest, compared with coolest, days of the year. The extent of this impact largely depended on the age of the infant. From approximately 4 months onwards, we predicted an increasingly negative impact of high

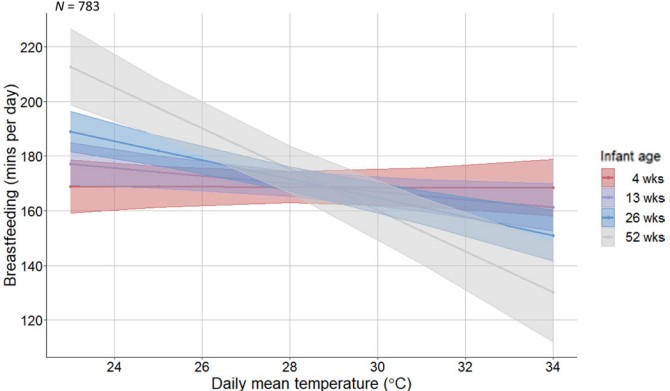

**Figure 4** Interaction effect of daily mean temperature (°C) and infant age on time spent breastfeeding (self-reported minutes/day). *N* denotes sample size.

temperature on breastfeeding duration. For younger infants, temperature had a lower impact. However, there was suggestive evidence that very young infants (under 3 months) were less likely to be breastfed exclusively as temperature increased.

There may be several explanations for the reduction in breastfeeding duration as temperatures rise in hot climates. Infants may demand less milk in order to limit heat generation, or they may become too uncomfortable to feed.[9] On the other hand, mothers may offer less breast milk due to their own discomfort under very hot conditions[9 47] and/or due to a misperception that babies require supplementary water, especially on hot days.[10 12 13]

Despite efforts to improve breastfeeding practices in sub-Saharan Africa,[48] the belief that breast milk is insufficient to hydrate babies and that water is needed to quench their thirst still prevails in several societies.[12 49–51] In the commune of Bobo-Dioulasso, provision of supplementary fluids (particularly water) was widespread. More than 95% of infants aged 3 months and older were given non-milk fluids in the 24 hours before interview. It is recommended that infants under 6 months be breastfed more often in hot weather,[52] and, ideally, the total intake of breast milk over a 24-hour period would increase to avoid infant dehydration. However, our findings tentatively suggest the contrary. Rather than increase breastfeeding to prevent infant dehydration, the women in our study may have provided a greater volume of supplementary fluids when temperatures increased.

The hot climate has been identified as a barrier to exclusive breastfeeding in the Democratic Republic of the Congo,[10] southern Zimbabwe,[13] Ghana[12] and Ethopia[49] and might (at least, partially) explain the very low rate of exclusive breastfeeding found herein. In the cooler subtropical climate of Bihar, India, odds of exclusive breastfeeding was significantly lower in summer than in winter or transitional seasons. Perceived thirst was proposed as an underlying cause for the higher rates of supplementary feeding in warmer months.[29] However, in contrast to our findings, the impact of season was greater for infants aged 3–6 months than for infants under 3 months.[29] The warmer climate of Bobo-Dioulasso, cultural values and beliefs around breastfeeding,[53] and the comparatively low rate of exclusive breastfeeding in our study (8% vs 70% of infants aged 3–6 months) likely explain this discrepancy in findings.

Many women across the world report perceived inadequacy of milk supply as their main reason for early weaning.[54] It is not clear if breast milk production is impacted by heat exposure, either directly or indirectly. High temperatures may exacerbate water stress, increasing the risk of dehydration among mothers in water-poor regions. However, the effects of maternal dehydration on breast milk production in hot climates are largely unknown.[55 56] Several studies have shown that exclusively breastfed infants maintain normal hydration under hot conditions,[57–59] indicating that the quantity of breast milk is not affected. However, field and experimental animal studies have shown that both the yield and nutritional composition of ruminant milk decline under hot conditions.[60] Further, the milk production capacity of animal mammary epithelial cells was found to decline following in vitro exposure to high temperatures (41°C).[61] To our knowledge, no studies have examined breast milk production in relation to temperature, but there is evidence that maternal stress affects breast milk composition[62] and delays secretory activation.[63] Even if milk supply is not adversely affected, high temperatures may contribute to women's perception of inadequate milk supply.

The marginal increase in exclusive childcare time with temperature is not easily explainable, given the range of activities included in this outcome (eg, bathing/dressing, playing/watching, and tending to when unwell). Tasks such as dressing children may take longer under hot conditions due to excessive sweating and/or low energy levels. Increased effort may be required to bathe or soothe children when temperatures rise, or women might spend more time monitoring other children in the household.

The main strengths of this study are the longitudinal dataset, population-based sampling, detailed questionnaire on activities, small loss to follow-up and extensive confounder control. The main limitations are the small sample size, which may have reduced our ability to detect statistically significant associations, and the short recruitment window (ideally, the study would have been conducted over several years). However, 2014 was a climatically typical year in Burkina Faso during 2010s.[25] We used meteorological data recorded at one weather station in Bobo-Dioulasso, located in the urban centre. We could not assign exposures to women's residential addresses, but daily variability in temperature exposures was likely to be consistent over the study area, even if absolute temperatures varied slightly.

Measurements of breastfeeding and childcare duration relied on self-reported time-use estimations. We put several measures in place to assist participants, including the recent and short (24-hour) recall period with questions aimed at establishing a 24-hour timeline. However, it is possible that temperature affected participants' ability to estimate time, despite the use of benchmarks (eg, path of the sun). Time-use diaries and direct observation offer arguably more robust methods for future research, although each has limitations. Our measurement of exclusive breastfeeding was not optimal, but women were not asked directly if they breastfed their infant exclusively. Instead, this outcome was constructed from women's recall of all fluids given to their child in the past 24 hours. Questions on infant feeding practices and childcare were embedded within an extensive interview schedule, further reducing the possibility of response bias. Finally, the outcome of childcare is complex and refers to time spent with children of all ages as this question was not specifically phrased to indicate the target (newborn) child.

Larger studies are needed to further examine the impacts of heat on infant feeding practices in hot

climates. Future research should consider temperature in relation to the number and duration of individual breast-feeds, and to the volume of breast milk and supplementary fluids consumed by infants over a 24-hour period. Research should also seek to determine if high temperatures impact on breast milk production. Actions should be taken to ensure that hot weather does not negatively impact on breastfeeding behaviour. Effective interventions are likely to require a multidimensional approach.[64] It is important that health workers and mothers are informed about normal heat-induced changes in infant breastfeeding patterns so that such changes are not misinterpreted as a need for supplementation.

## CONCLUSIONS

Exclusive breastfeeding is an essential cornerstone for the well-being and survival of infants. Our findings suggest a substantial decrease in breastfeeding duration, and potentially lower odds of exclusive breastfeeding very young infants, during hot weather. These findings are important as infants require increased hydration to cope physiologically with increased heat, and the safest form of hydration for young infants is breast milk.

Larger studies are needed in Burkina Faso and beyond as climate change in Africa is accelerating.[65] Without effective interventions, mothers may find it increasingly difficult to breastfeed their infants as temperatures rise. Maternal and child health programmes in hot climates should be updated to improve messaging and breastfeeding practices during extreme hot weather.

## Author affiliations
[1]Department of Public Health, Environments and Society, London School of Hygiene & Tropical Medicine, London, UK
[2]Department of Infectious Disease Epidemiology, London School of Hygiene & Tropical Medicine, London, UK
[3]Agence de Formation de Recherche et d'Expertise en Santé pour l'Afrique (AFRICSanté), Bobo-Dioulasso, Burkina Faso
[4]Division of Child and Adolescent Health, Institute of Clinical Medicine, University of Oslo, Oslo, Norway
[5]Department of Pediatrics and Adolescent Health, University of Botswana, Gaborone, Botswana
[6]Department of Medicine, Clinical Epidemiology Division, Karolinska Institutet, Stockholm, Sweden
[7]Departement Biomédical et Santé Publique, Institut de Recherche en Sciences de la Santé, Ouagadougou, Burkina Faso
[8]Wits Reproductive Health and HIV Institute, University of the Witwatersrand, Johannesburg, South Africa
[9]Department of Population Health, Medical College, Aga Khan University, Nairobi, Kenya

**Twitter** Véronique Filippi @1verofilippi

**Acknowledgements** We thank Maurice Yaogo, Patrick Ilbouldo, André Soubeiga, Katerini Storeng, Denis Ouedraogo, Seydou Drabo and Tim Powell-Jackson for their valuable contribution to the primary (PopDev) study. We gratefully acknowledge all women who participated in the PopDev study. We also thank two peer-reviewers for their thoughtful and insightful comments, which improved the final manuscript.

**Contributors** VF and SKov conceived the study (secondary analysis). JAC and RG collected data. CP conducted the statistical analysis under the supervision of SH, and drafted the paper. All authors (CP, VF, JAC, RG, SH, BN, NR, KK, MC, AL, SKou and SKov) interpreted the results, critically revised the paper for important intellectual content, approved the final version for publication, and agreed to be accountable for all aspects of the work. CP is responsible for the overall content as guarantor.

**Funding** The PopDev study was supported by the Economic and Social Research Council (ESRC) in response to the Joint ESRC-WOTRO-RCN-PRB-Hewlett call on Population & Development (grant number ES/K011049/1). The secondary analysis was supported by the Natural Environment Research Council (grant numbers NE/T013613/1 and NE/T01363X/1), the Research Council of Norway (grant number 312601) and The Swedish Research Council for Health, Working Life and Welfare in collaboration with the Swedish Research Council (Forte) (grant number 2019-01570), coordinated through a Belmont Forum partnership.

**Competing interests** None declared.

**Patient and public involvement** Patients and/or the public were involved in the design, conduct, reporting or dissemination plans of this research. Refer to the Methods section for further details.

**Patient consent for publication** Not applicable.

**Ethics approval** This study involves human participants and was approved by the research ethics committees of Centre Muraz, Burkina Faso (ref: A16-2013/CE-CM), and the London School of Hygiene & Tropical Medicine (ref: 6401). The participants gave informed consent to participate in the study before taking part.

**Provenance and peer review** Not commissioned; externally peer reviewed.

**Data availability statement** Data are available upon reasonable request. Individual-level deidentified participant data from the primary (PopDev) study are available to researchers who have a valid research question, which is not being investigated by the primary research team. A data sharing agreement will be required. All data requests should be made via https://datacompass.lshtm.ac.uk/id/eprint/64/. The questionnaires, consent form, data dictionary (codebook) and user guide are publicly available from https://datacompass.lshtm.ac.uk/id/eprint/64/. Data have been available from 2018 with no end date. Meteorological data are publicly available from TuTiempo (https://www.tutiempo.net/). Statistical code is available from the corresponding author.

**ORCID iDs**
Chérie Part http://orcid.org/0000-0002-3281-1671
Nathalie Roos http://orcid.org/0000-0001-9752-2355
Adelaide Lusambili http://orcid.org/0000-0001-8174-7963

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
