## [Reviewer comments · BMJ Open]

ARTICLE DETAILS

TITLE (PROVISIONAL)	How do high ambient temperatures affect infant feeding practices? A prospective cohort study of postpartum women in Bobo-Dioulasso, Burkina Faso
AUTHORS	Part, Chérie; Filippi, Veronique; Cresswell, Jenny; Ganaba, Rasmané; Hajat, Shakoor; Nakstad, Britt; Roos, Nathalie; Kadio, Kadidiatou; Chersich, Matthew; Lusambili, Adelaide; Kouanda, Seni; Kovats, Sari

VERSION 1 – REVIEW

REVIEWER	Cliffer, Ilana Harvard University T H Chan School of Public Health, Global Health and Population
REVIEW RETURNED	08-Apr-2022

GENERAL COMMENTS	General comments: This is a very interesting and well-done analysis with important implications for infant and young child feeding in the context of a changing climate. The genesis of the study questions having come out of focus groups with pregnant and postpartum mothers ensure that the topic is relevant and important to the community from which the data are derived. The analyses are well-explained overall and the statistics used are appropriate, but could use more detail about modeling decisions and procedures in the methods section to allow for full replicability of the study. A few specific suggestions follow:  [ ] Abstract  o Participants: clarify who is the subset of participants in this secondary analysis. o Since this is a cohort, also state briefly how many follow-ups were done at what time points? At least those that are relevant to this study. o Exposure: name the archive where the temperature data were downloaded from o Primary outcome measures: supplementary feeding of infants < 6 months should be reported separately than those 6-12 months [ ] Methods  o Participants:  [ ] Was the 2006 census used as the sampling frame to select locality clusters? What was used to select individuals? [ ] Please clarify what were the locality clusters where women were recruited from. Clinics? Households? [ ] Clarify if this is a subset of participants in this secondary analysis. The authors mention that participants were between 20 weeks gestation and 22 weeks postpartum, but you focus only on postpartum women who are feeding infants. Please discuss
--

	whether all women in the original sample are also included in this secondary analysis, or if this is a subset of those who completed all three interviews post-partum?  [ ] Table 1 includes the % of women postpartum at each interview round, which is very helpful for understanding who the sample of women is, but it would be good to give readers a better understanding of this early on, perhaps in the methods section, to frame the study. Better description of who the participants in the secondary analysis are might accomplish this.  o Data collection  [ ] Were any devices/strategies used to help women with time estimations? Time estimation can be very tricky and concepts of time can change based on contexts. Please describe how enumerators worked with participants to ensure a mutual understanding of the notion of time, and ensure consistency among all study participants in terms of estimation of minutes spent on activities. Self-reported time should also be mentioned and discussed as a limitation (self-reports are discussed, but specifically that time is very hard to estimate). [ ] Lines 143-146: It is a little bit unclear exactly how many interviews were conducted per participant. My understanding was that it was only 3 total, but this section makes it seem like there may have been three post-delivery. Please make sure this is clear. o Meteorological data: what was the range of distance of study individuals from the weather station? o Data analyses  [ ] Overall, the data analysis section needs to be filled out more, to better and more specifically describe the modeling methods used and decisions made. Some of these become apparent in the results section based on how the results are presented, but they should be made very clear in the methods section. [ ] Lines 171-176: could add a phrase up front to state that data for supplementary feeding and exclusive breastfeeding were analyzed cross-sectionally at a single time point? [ ] Since optimal supplementary feeding practices and exclusive breastfeeding are different among infants ≥ 6 months or < 6 months, I would expect the models for these two outcomes to either be stratified by age or to use an indicator for “optimal supplementary feeding” as the outcome in which optimal feeding would mean something different for infants ≥ 6 months or < 6 months. [ ] After reading the results section, I can tell that supplementary feeding models were adjusted for age. I would suggest stratifying by age instead of adjusting for it, since interpretation of the results is inherently different for children ≥ 6 months or < 6 months. [ ] I see in the results section that models for EBF were restricted to infants < 6 months – please state in the methods section that this was done. [ ] Line 194: where did data on apparent daily mean temperature come from? [ ] Lines 197-198: do the authors mean that disaggregating by child sex was also not possible since this information was not collected in the primary study? Clarify what is meant by gender-disaggregated, and how that is different from the previous phrase which talks about sex-disaggregated. [ ] I see based on the results section that long-term trends were adjusted for, but please make clear how this was done, in the methods section. o Results  o Table 1
--	---

	 [ ] Please specify in a footnote that the units for continuous variables are self-reported (minutes/day). [ ] Are medians and IQRs used in Table 1 for continuous variables because they are not normally distributed? Perhaps specify, if so, or explain choice of median and IQR instead of mean and SD otherwise. o It seems that a lot of the results are not presented in tables, rather they are only described in the text. This may be a personal preference, but I find it much easier to read and interpret results if they are also laid out in tables that I can reference. Please consider adding tables to report descriptive stats for all outcomes and covariates in one place, and stratifying this table by infant age. o It would be good to see the breakdown of mean infant age by calendar month of interview as well, to allow readers to assess how overall time trends may impact interpretation of results. o Figures – please provide sample sizes for all figures, and for Figure 3, stratified sample sizes. o Can confidence intervals be added to Figure 3? [ ] Discussion o In the limitations section, the authors could elaborate on how self-reported time estimations may have influenced findings, and next steps for measuring this more robustly (direct observation?).
--	--

REVIEWER	Grace, Kathryn University of Minnesota Twin Cities
REVIEW RETURNED	31-May-2022

GENERAL COMMENTS	In this article the authors attempt to investigate a series of important questions about how temperature (especially heat) may impact infant care practices in a community (and surrounding areas) in Burkina Faso. Overall the article is well written and brings up an important topic - how women's lives are impacted in the context of climate change. Thank you for the chance to read it and reflect on this important and understudied topic. There are some important issues that I think the authors need to address to strengthen the contribution of this article. The background section that is designed to provide some insight into why temperature is linked to the four outcome variables (self-reported breastfeeding duration, exclusive breastfeeding, supplementary feeding, and childcare duration on the day before interview), was actually not well described. For example, the question of childcare and why that would be related to temperature was not well justified (similarly there is no connection described in the conclusion). There is some literature on the topic that the authors cite but it primarily summarizes the connection between temperature/rainfall and these different dimensions of infant care through the food security and/or agricultural labor pathway. This approach does not seem relevant here since the authors are deliberately not investigating the food security pathway. I hope that the authors will offer a much clearer link between daily seasonal temperature differences (e.g., not extreme temps) and also the different key variables as there is a lot to be clarified here. Additionally, the authors cite work which presumably comes from a project associated with theirs - citation #7 to support their approach and justify their findings. This citation seems to reflect a qualitative study conducted in 2020 (or perhaps an analysis of qualitative work gathered from an earlier time period). I was not able to find this article through an online search of the key
---

academic search engines and it's not clear to me if the paper was actually peer reviewed. It's always okay by me if authors cite their own work, but in this case it seems like they are using this piece which is not available for a reader to access, as an important piece of evidence to support their framing and their findings. I encourage the authors to provide more evidence of this work and somehow include it in this text but at the very least they need to provide additional citations of peer-reviewed literature to help strengthen their argument and justify their findings. They do provide additional citations and text in the discussion and some of that would really bolster the background section at the front of the paper.

I would have also liked to see more discussion of the setting (is it possible to provide a map of the surveyed areas?). It was not clear to me how proximate the survey data points are to each other (I'm unclear about what it means to be in a rural area in Bobo) or how proximate the data are to the temperature station (temp does spatially vary).

Relatedly, the authors briefly mention the source of the data - the PopDev program but they do not provide enough details on the data collection. Were the data collected in French (meaning only French speaking participants)? How long did the surveys take? And, perhaps most importantly, how are these kinds of measurements (minutes of breastfeeding or time spent caring for children, for example) collected and validated? Technically, it seems as though the question asks about how many minutes was a child "given the breast" - is this a culturally appropriate way of capturing breastfeeding duration? Have these measures been used elsewhere? I was not able to find this information when I went through the website for the data either. (Also, as a side note, the authors say that the list of tasks is "exhaustive" but when I looked at the survey it is not really exhaustive, but rather a relatively short list of questions about time use.). Because these measures are so key, it seems important that the authors spend more time addressing the details of the measures as well as address some of these issues in the limitations/conclusions sections.

More citations and supporting text on what childcare means in this setting is vital - especially because this is sensitive to temperature and the authors do not really prepare us for why that might be with regard to what is actually being measured in this context.

The authors do address some of the issues around breastfeeding duration in the conclusion (e.g., the address the issue of fore vs. hindmilk). This might be helpful to also address in the beginning of the paper.

In terms of the temperature measurements, it was not clear to me why the authors used mean temp and not temp max or humidity (to get a "feels like"). The authors also did not provide any justification for their temp measurement either. Considering alternative measures of temperature might be useful here. (a small side note is that the authors do sometimes mention "heat stress" in the paper but really they are not measuring heat stress - rather they are measuring temperature and their results may have implications for how we think about heat stress, but heat stress is an individual-level biological process which is not measured here - please be careful on the use of the language around heat stress in

	the paper. The authors also mention that 32.5 c is "very hot" - it seems like that is relative - especially as compared to the north of Burkina). In terms of the results, I would have liked to see more discussion about how variable the temperature data actually was. It would be nice to know more about the distribution of the temperature data used in the analysis. Perhaps more discussion of Fig 3 discussing the difference (by infant age group) in duration for temps at the low and high end of the spectrum (with some indication of the count of observations). I generally think a discussion around the range of temps in the study and the associated outcomes would help contextualize the findings. The discussion of the slope (e.g., change in duration associated with a 1 C increase) is less interesting than a discussion of the differences at the high and low end. I don't mean a discussion of the "hottest and coolest days" as in line 317 but more within a single wave of data collection (or among kids about the same age) how much variability did you really have in temp. I found it interesting that women with children under 6 months old did not really change their exclusive breastfeeding practices, even as temperatures increased and think the authors should discuss the underlying reasons they think may exist for this (as relates to their literature review). The authors mostly focused on the duration of the breastfeeding findings and a little around the supplementary feeding findings but some of the other findings are quite interesting as well. For the limitations section - is there any possibility of addressing anything like co-nursing or other adults in the house who might help with care/nursing? I'm not sure if this is relevant in this setting. I also wonder about the use of formula - is it possible that individuals are using formula? That seems like quite a difference in health impacts as opposed to something like water. In other words - there are good alternatives to breastfeeding that women may rely on and not all supplements should be considered as equal. Again, I'm not sure if this is relevant in this setting but it seems that women should always have access to high quality alternatives to breastfeeding as part of ensuring women's autonomy and equality.
--	--

VERSION 1 – AUTHOR RESPONSE

Reviewer #1.

Dr. Ilana Cliffer, Harvard T.H. Chan School of Public Health, Harvard University

General comments: This is a very interesting and well-done analysis with important implications for infant and young child feeding in the context of a changing climate. The genesis of the study questions having come out of focus groups with pregnant and postpartum mothers ensure that the topic is relevant and important to the community from which the data are derived. The analyses are well-explained overall and the statistics used are appropriate, but could use more detail about modeling decisions and procedures in the methods section to allow for full replicability of the study.

Thank you for your positive comments. We now include additional detail on our modelling decisions and procedures in the methods section, as highlighted in response to your specific suggestions below.

To clarify, the genesis of the research questions for the secondary analysis came from qualitative fieldwork conducted in Burkina Faso (in-depth interviews and focus group discussions) by the CHAMNHA project. The genesis for the initial (PopDev) study was different, as it followed on from previous mixed method research conducted in Burkina Faso.

A few specific suggestions follow:

Abstract

- **Participants: clarify who is the subset of participants in this secondary analysis.**

We have clarified that the secondary analysis focused on postpartum women (line 34):

“Our secondary analysis focused on postpartum women (n=857).”

- **Since this is a cohort, also state briefly how many follow-ups were done at what time points? At least those that are relevant to this study.**

We now include the following (lines 31-34):

“Participants were interviewed at three time points: cohort entry (when between 20 weeks gestation and 22 weeks postpartum), three and nine months thereafter. Retention at nine-month follow-up was 90%.”

- **Exposure: name the archive where the temperature data were downloaded from.**

Thank you for pointing this out. We now name the archive “*(TuTiempo.net)*” where the temperature data were downloaded from (line 36).

- **Primary outcome measures: supplementary feeding of infants < 6 months should be reported separately than those 6-12 months**

We agree with your comment. We had originally examined exclusive breastfeeding of infants < 6 months, and supplementary feeding of infants aged 0-12 months (adjusting for age). We have now redefined and reanalysed our primary outcome measure of supplementary feeding to focus only on infants aged 6-12 months. Supplementary feeding of infants < 6 months was inversely assessed by our exclusive breastfeeding outcome (“no fluids other than breast milk provided [to infants under 6 months] in past 24 hours”).

Methods

- **Participants:**
 - **Was the 2006 census used as the sampling frame to select locality clusters? What was used to select individuals?**

Yes, the 2006 census was used as the sampling frame to select locality clusters. Households were visited per selected cluster, using a modification of the WHO’s Expanded Program on Immunization sampling methodology, to identify 30 women (between 7 months gestation and 3 months postpartum) per cluster. It was calculated that each cluster needed a minimum of 300-330 households to identify 30 pregnant/postpartum women. We have added the following to our ‘Participants’ subsection (lines 131-136; new text in *italics*):

“The 2006 census was used as the sampling frame to select locality clusters. It was estimated that each cluster must contain a minimum of 300-330 households to identify 30 eligible participants per cluster. Thirty-eight locality clusters were identified (14 urban, 24 rural), and participants were recruited within households at cluster-level. Households were visited per selected cluster using a modification of the World Health Organization’s Expanded Program on Immunization sampling methodology.”

- **Please clarify what were the locality clusters where women were recruited from. Clinics? Households?**

Women were recruited within households at the cluster level. We have included this information on line 134 (see above).

- **Clarify if this is a subset of participants in this secondary analysis. The authors mention that participants were between 20 weeks gestation and 22 weeks postpartum, but you focus only on postpartum women who are feeding infants. Please discuss whether all women in the original sample are also included in this secondary analysis, or if this is a subset of those who completed all three interviews post-partum?**

We have now clarified those participants included in the secondary analysis (lines 141-142):

“All women who completed at least one interview postpartum (n=857) were included in the secondary analysis (see figure 1).”

- **Table 1 includes the % of women postpartum at each interview round, which is very helpful for understanding who the sample of women is, but it would be good to give readers a better understanding of this early on, perhaps in the methods section, to frame the study. Better description of who the participants in the secondary analysis are might accomplish this.**

Thank you for this comment. Following on from the comment above, we have included a flow chart in the methods section to give readers a better understanding of the study sample and data used in the secondary analysis:

“Figure 1. Sampling flow chart for secondary analysis, showing number of interviews conducted with pregnant (blue) and postpartum women (green) at each interview round (T). Data from all interviews highlighted in green were included in the secondary analyses.”

- **Data collection**

- **Were any devices/strategies used to help women with time estimations? Time estimation can be very tricky and concepts of time can change based on contexts. Please describe how enumerators worked with participants to ensure a mutual understanding of the notion of time, and ensure consistency among all study participants in terms of estimation of minutes spent on activities. Self-reported time should also be mentioned and discussed as a limitation (self-reports are discussed, but specifically that time is very hard to estimate).**

Thank you for this comment. Indeed, time estimation can be very tricky. Therefore, a strategy was developed and agreed during interviewer training to assist participants in estimating their time spent on each activity. Participants were first asked to list all activities that they had engaged in, by time of day. After listing the activities during a given period of time, the investigator proceeded with more in-depth questions in order to help participants to estimate the duration of each activity. Here, indications of time (particularly the path of the sun), combined with current events of life, served as benchmarks. Once participants had finalised their estimates of minutes spent on each activity during the said period, the interviewer and participant repeated the estimation process for the following time period, until the full 24-hour recall period was complete. It should be noted that each participant had a notebook in which time-use estimates were drafted and revised before final responses were recorded in the questionnaire. We now describe this process in the 'Data collection' subsection of our methods (lines 166-187):

"The recall period was defined as, "between waking up yesterday morning and waking up this morning". Participants described their activities and the interviewer categorised them using a pre-defined list, which was added to when needed. Breastfeeding, caring for children (bathing/dressing, feeding, playing/watching, tending to when unwell), income-generating work (e.g. agro-processing for trade, sale of products at market, small business, office activities), attending classes (e.g. literacy courses), and household chores (e.g. preparing meals, cleaning clothes, washing dishes, fetching water, fetching fuel) were included in this list.

To assist participants in time-use recall, women were initially asked to list all activities that they had engaged in during a specific time of day (e.g. between waking up and midday). Participants were then asked probing questions to assist them in estimating the duration of each activity. For example, some women said that they woke with the call of the muezzin, which enabled the interviewer to determine time of wake. Participants would then be asked if they began their first listed activity immediately, or if they did something else first. The interviewer asked when their first activity ended to which women responded, "to the first rays of sunshine", for example. Thus, the interviewer had adequate information to estimate time duration of the first activity. Once participants had finalised their time-use estimates for the specified period, the process was repeated for the next time of day, until the full 24-hour recall period was complete. Participants used a notebook to draft and revise time-use estimates before final responses were recorded in the questionnaire. Indications of time (particularly the path of the sun), combined with current events of life, served as benchmarks for time estimations."

We have also added and discussed self-reported time-use as a study limitation (lines 449-454):

"Measurements of breastfeeding and childcare duration relied on self-reported time-use estimations. We put several measures in place to assist participants, including the recent and short (24-hour) recall period with questions aimed at establishing a 24-hour timeline.

However, it is possible that temperature affected participants' ability to estimate time, despite the use of benchmarks (e.g. path of the sun). Time-use diaries and direct observation offer arguably more robust methods for future research, although each has limitations."

- **Lines 143-146: It is a little bit unclear exactly how many interviews were conducted per participant. My understanding was that it was only 3 total, but this section makes it seem like there may have been three post-delivery. Please make sure this is clear.**

We aimed for 3 interviews per participant. Almost 50% of women were postpartum at cohort entry. Therefore, when not lost to follow-up, these women were interviewed three times post-delivery. Women who were pregnant at cohort entry were interviewed two times post-delivery, when not lost to follow-up. We hope that the above flowchart now clarifies this for the reader.

We have also added to lines 143-146 (now lines 188-189) to further clarify:

"At first interview following childbirth (interview round one or two), participants were asked how many children were born and the date of delivery. At interview rounds two and three ..."

- **Meteorological data: what was the range of distance of study individuals from the weather station?**

GPS data were not collected and residential addresses were not available for each participant. The weather station was located in the industrial district of the large urban centre (city of Bobo-Dioulasso). All participants lived within 40-50 km of the urban centre. We have included this information in the 'Setting' subsection of our methods (lines 147-148):

"All rural participants in this study resided within 40-50 km of the city."

We have also included this as a limitation in our discussion (lines 445-448):

"We used meteorological data recorded at one weather station in Bobo-Dioulasso, located in the urban centre. We could not assign exposures to women's residential addresses, but daily variability in temperature exposures were likely to be consistent over the study area, even if absolute temperatures varied slightly."

- **Data analyses**

- **Overall, the data analysis section needs to be filled out more, to better and more specifically describe the modeling methods used and decisions made. Some of these become apparent in the results section based on how the results are presented, but they should be made very clear in the methods section.**

We have added detail to the 'Data analysis' subsection of our methods to better describe the modelling methods used and decisions made. For example:

"Daily exposures were linked with outcomes at individual level, by date of interview minus one-day (t-1) to reflect same-day temperature when activities were undertaken."

"Categorical outcomes, potential confounding variables and covariates (see table S1) were summarised as proportions (expressed as percentages). Continuous variables were summarised as mean \pm standard deviation (SD) if normally distributed, or as median and interquartile range if not normally distributed. Summary statistics were stratified by interview round where applicable."

“Multilevel linear regression was used to estimate the effects of daily mean temperature on time spent (i) breastfeeding, and (ii) caring for children. *This approach made use of all available time-use data on breastfeeding and childcare, while accounting for the longitudinal and nested structure of the data.*”

“Each level was defined as a random coefficient with random intercept *to allow for correlation within-individuals and clusters. A first-order autoregressive correlation structure allowed for unequal spacing of interviews.*”

“*Indicator terms were included for calendar month of interview to adjust for season and long-term trends.*”

“*Other covariates (number born [singleton; multiple birth], infant age [weeks], maternal age [≤ 19 ; 20–34; ≥ 35 years], gravidity [1; 2–5; ≥ 6 pregnancies], residential area [urban; rural], living arrangements [with partner full-time; with partner periodically; not with partner; not in a relationship], paid work or education [minutes/day], domestic work [minutes/day], and roofing materials [natural; rudimentary; contemporary]; see table S1) were added to the models one-by-one, following a forward stepwise process.*”

Also see responses below.

- **Lines 171-176: could add a phrase up front to state that data for supplementary feeding and exclusive breastfeeding were analyzed cross-sectionally at a single time point?**

We included a statement to that effect in our original manuscript, but we have added text to clarify the approach taken and our reasoning behind it (lines 259-263):

“*Data were available for two time points (interview rounds two and three). To reduce model complexity, binary outcomes were analysed cross-sectionally, at single time points: Exclusive breastfeeding at interview round two, and supplementary feeding at interview round three, based on the age range of infants at each round.*”

- **Since optimal supplementary feeding practices and exclusive breastfeeding are different among infants ≥ 6 months or < 6 months, I would expect the models for these two outcomes to either be stratified by age or to use an indicator for “optimal supplementary feeding” as the outcome in which optimal feeding would mean something different for infants ≥ 6 months or < 6 months.**

This is a good point. Given WHO’s (2021) recommendations that infants < 6 months are breastfed exclusively, and that supplementary foods are only introduced ≥ 6 months, we restricted our analysis of exclusive breastfeeding to infants < 6 months and our analysis of supplementary feeding to infants aged 6-12 months. The definitions of exclusive breastfeeding and supplementary feeding used were inversely proportional, therefore we did not consider exclusive breastfeeding of infants 6-12 months, nor supplementary feeding of infants < 6 months. However, considering that exclusive breastfeeding of infants < 6 months is uncommon in Burkina Faso (Issaka et al., 2017) and rates may decrease as infants age (Das et al., 2016), we stratified exclusive breastfeeding by infant age (< 3 months; 3 to < 6 months). We added the following text to our paper (lines 256-259 and 267-269):

“*We restricted our analyses of exclusive breastfeeding and supplementary feeding to infants aged less than six months and 6-12 months, respectively. This follows from WHO’s recommendations that infants are breastfed exclusively for the first six months of life, and that supplementary foods are only introduced thereafter. [17]*”

“As the effects of temperature on exclusive breastfeeding may change as infants age, [30] our exclusive breastfeeding analysis was age-stratified (< 3 months; 3 to < 6 months).”

- **After reading the results section, I can tell that supplementary feeding models were adjusted for age. I would suggest stratifying by age instead of adjusting for it, since interpretation of the results is inherently different for children ≥ 6 months or < 6 months.**

We agree with your comment, and have amended our analyses accordingly, as described above.

- **I see in the results section that models for EBF were restricted to infants < 6 months – please state in the methods section that this was done.**

This has now been stated in our methods section (lines 256-257), as described above.

- **Line 194: where did data on apparent daily mean temperature come from?**

Apparent daily mean temperature was calculated from the meteorological data on temperature, relative humidity, and wind speed, using a function within the HeatStress package in R. We now present this information earlier, in the ‘Exposure’ subsection of our methods (lines 219-221):

“Apparent daily mean temperature ($^{\circ}\text{C}$) was calculated from daily mean temperature, relative humidity and windspeed, using the R HeatStress package, [41] to test the robustness of our findings.”

- **Lines 197-198: do the authors mean that disaggregating by child sex was also not possible since this information was not collected in the primary study? Clarify what is meant by gender-disaggregated, and how that is different from the previous phrase which talks about sex-disaggregated.**

This statement was included in our manuscript in accordance with the Sex and Gender Equity in Research (SAGER) guidelines

(<https://researchintegrityjournal.biomedcentral.com/articles/10.1186/s41073-016-0007-6>).

However, we agree that this statement may confuse the reader and we have therefore removed this text from the manuscript.

- **I see based on the results section that long-term trends were adjusted for, but please make clear how this was done, in the methods section.**

We included indicator terms for calendar month of interview to adjust for season and long-term trends (lines 243-244):

“Indicator terms were included for calendar month of interview to adjust for season and long-term trends.”

Results

• Table 1

- **Please specify in a footnote that the units for continuous variables are self-reported (minutes/day).**

We had previously specified the units for time use data in table 1 as ‘Time use (mins/day)’. We have now amended this to ‘Time use (self-reported minutes/day)’.

- **Are medians and IQRs used in Table 1 for continuous variables because they are not normally distributed? Perhaps specify, if so, or explain choice of median and IQR instead of mean and SD otherwise.**

Yes, we presented medians and IQRs in table 1 for continuous variables because they were not normally distributed, at least when stratified by interview round. We have added text to the 'Data analysis' section of our methods to justify our use of mean \pm SD or median and IQR (lines 224-228), as noted above.

- **It seems that a lot of the results are not presented in tables, rather they are only described in the text. This may be a personal preference, but I find it much easier to read and interpret results if they are also laid out in tables that I can reference. Please consider adding tables to report descriptive stats for all outcomes and covariates in one place, and stratifying this table by infant age.**

Thank you for your comment. Given the original cohort study design, we believe that it is important to provide the readers with descriptive statistics stratified by interview round. We have included descriptive stats for most outcomes and covariates in table 1. It is difficult to report descriptive statistics for all covariates in a single table as some (for example, number of infants born, maternal age, and roof materials) were collected at a single time point. Thus, the same covariates would not be included in a table stratified by infant age, as you suggest. We therefore supplemented the information in table 1 with additional information in text.

- **It would be good to see the breakdown of mean infant age by calendar month of interview as well, to allow readers to assess how overall time trends may impact interpretation of results.**

Overall time trends should not impact on interpretation of results as our models adjusted for calendar month of interview (and infant age where appropriate). We have provided descriptive stats (median and IQR) for infant age by interview round in table 1 to summarise the change in infant age over time.

- **Figures – please provide sample sizes for all figures, and for Figure 3, stratified sample sizes.**

We have added sample sizes to all figures. Figure 3 (now figure 4) does not present results of a stratified analysis, but rather the interaction effect of temperature and infant age on breastfeeding duration. We have now clarified this in the figure legend (lines 697-698):

“Interaction effect of daily mean temperature (°C) and infant age on time spent breastfeeding (self-reported minutes/day). N = sample size”

- **Can confidence intervals be added to Figure 3?**

Thank you for this suggestion. We have added confidence intervals to Figure 3 (now 4).

Discussion

- **In the limitations section, the authors could elaborate on how self-reported time estimations may have influenced findings, and next steps for measuring this more robustly (direct observation?).**

We have added to our discussion around the limitation of time-use estimations (lines 452-454):

“However, it is possible that temperature affected participants’ ability to estimate time, despite the use of benchmarks (e.g. path of the sun). Time-use diaries and direct observation offer arguably more robust methods for future research, although each has limitations.”

Reviewer #2

Kathryn Grace, University of Minnesota Twin Cities

Comments to the Author:

In this article the authors attempt to investigate a series of important questions about how temperature (especially heat) may impact infant care practices in a community (and surrounding areas) in Burkina Faso. Overall the article is well written and brings up an important topic - how women's lives are impacted in the context of climate change. Thank you for the chance to read it and reflect on this important and understudied topic.

Thank you for your positive comments.

There are some important issues that I think the authors need to address to strengthen the contribution of this article.

- 1. The background section that is designed to provide some insight into why temperature is linked to the four outcome variables (self-reported breastfeeding duration, exclusive breastfeeding, supplementary feeding, and childcare duration on the day before interview), was actually not well described. For example, the question of childcare and why that would be related to temperature was not well justified (similarly there is no connection described in the conclusion). There is some literature on the topic that the authors cite but it primarily summarizes the connection between temperature/rainfall and these different dimensions of infant care through the food security and/or agricultural labor pathway. This approach does not seem relevant here since the authors are deliberately not investigating the food security pathway.**

Thank you for pointing this out. We included the Randell et al. (2021) study that the reviewer mentions as this was one of the only quantitative studies we found that set out to examine the impacts of weather variables on exclusive breastfeeding. We now remove the detail of this study and better introduce the possible links between temperature and our study outcomes (lines 67-74 and 85-104):

“High temperatures may also reduce cognitive function [7] and interfere with daily activities, leading to a decline in emotional health and wellbeing. [8] Mothers may find it difficult to breastfeed their infants under extreme heat, [9] and may also change their behaviour due to perceived risks to health. For example, there is still a common misconception among postpartum women in several African countries that breast milk is not sufficient to hydrate babies during hot weather; leading to supplementary feeding of infants, with sometimes not potable water, [10-13] and a reduction in exclusive breastfeeding. [10]”

“It is not unusual for breastfeeding patterns to change in hot weather. Infants may refuse to feed during the hottest part of the day, or they may demand more frequent, but shorter, feeds throughout the day. [19] In doing so, babies consume mostly low-fat milk (foremilk) and avoid breast milk with a high fat content (i.e. afternoon/evening milk and hindmilk). [19, 20] Mothers must change their breastfeeding patterns to accommodate their infants’ needs, and may spend more time breastfeeding as temperatures rise. Conversely, women may spend less time breastfeeding during periods of high temperature due to increased discomfort for both mother and child, [9] increased provision of water

(believed necessary to quench baby's thirst in some African settings), [21] and/or associated health effects, such as low energy [8] and heat exhaustion. [22]

Infants and young children are particularly vulnerable to heat injury and dehydration due to a greater surface area to body mass ratio. [24] Therefore, as temperatures rise in hot climates, mothers may spend more time watching over their children and other children in the household, keeping them hydrated, and tending to them when unwell. Such increased demands on time may cause difficulties for mothers in low-income countries such as Burkina Faso, where women work to supplement household income (particularly in agriculture, horticulture, and small trade) as well as undertaking important domestic responsibilities (including gathering food, water, fuel, and feeding livestock). [6] Most women work in the informal sector, [25] therefore paid maternity leave is uncommon and many women return to work early in the postpartum period."

We now describe the connection between temperature and childcare in our discussion (lines 434-439):

"The marginal increase in exclusive childcare time with temperature is not easily explainable given the range of activities included in this outcome (e.g. bathing/dressing, playing/watching, tending to when unwell). Tasks, such as dressing children, may take longer under hot conditions due to excessive sweating and/or low energy levels. Increased effort may be required to bathe or soothe children when temperatures rise, or women might spend more time monitoring other children in the household."

2. I hope that the authors will offer a much clearer link between daily seasonal temperature differences (e.g., not extreme temps) and also the different key variables as there is a lot to be clarified here.

We are confident that there is very little literature on this topic, particularly quantitative studies, or qualitative studies that set out specifically to examine the effects of temperature. None that we know of examine daily temperature differences in relation to the key variables in our study. At best, most studies report seasonal differences in breastfeeding behaviours, which is insufficient to demonstrate a clear link with temperature as competing demands on women's time (e.g. agricultural and domestic work) and other factors (e.g. household food security) also change with season. Therefore, we have moved some text from the discussion to highlight this issue (lines 105-113):

"Average monthly temperatures in Burkina Faso range between 25–33°C [26] and the impacts on infant care practices are largely unknown. Studies in South America, South Asia and Africa show seasonal differences in breastfeeding behaviour, with conflicting results. [27-30] For example, in Bihar, India, infants under six months were more likely to be exclusively breastfed in the cooler than warmer season. [30] Whereas, in rural Egypt, exclusive breastfeeding of infants aged 6–11 months was more prevalent in the hot than cool season. [27] However, such studies are not sufficient to demonstrate an effect of ambient temperature as the competing time demands of women's domestic and agricultural workloads, [10, 31-34] as well as other potentially important drivers (e.g. household food security), also vary with season and weather in rural settings. [34, 35] With daily temperatures in West Africa expected to exceed 50°C in some regions, [36] further research is essential so that maternal and child health programmes can be updated."

3. Additionally, the authors cite work which presumably comes from a project associated with theirs - citation #7 to support their approach and justify their findings. This citation seems to reflect a qualitative study conducted in 2020 (or perhaps an analysis of qualitative work gathered from an earlier time period). I was not able to find this article through an online search of the key academic search engines and it's not clear to me if the paper was actually peer reviewed. It's always okay by me if authors cite their own work, but in this case it seems like they are using this piece which is not available for a reader to access, as an important piece of evidence to support their framing and their findings. I

encourage the authors to provide more evidence of this work and somehow include it in this text but at the very least they need to provide additional citations of peer-reviewed literature to help strengthen their argument and justify their findings.

Unfortunately, this is an understudied topic and we are confident that there is very little relevant literature beyond that now cited in our manuscript. As you rightly point out, citation #7 (now #9) relates to qualitative work undertaken within our project in 2020, which is currently in prep. for publication. This work informed our objectives for this secondary analysis and is described in the patient and public involvement section (lines 295-302):

“Pregnant and postpartum women, as well as community members, in the Kaya and Bogodogo health districts of Burkina Faso were involved before the secondary study began. In-depth interviews with pregnant and postpartum women (n=40), and focus group discussions with community members, were undertaken in October–November 2020. [9] The objectives for the secondary analysis were developed and informed by the lived experiences of postpartum women reported during this qualitative work. Specifically, women described how hot weather impedes breastfeeding due to excessive sweating and the discomfort of both mothers and their babies. [9]”

The reference refers readers to a Special Issue of Tropical Medicine and International Health: Abstracts of the 12th European Congress on Tropical Medicine and International Health, 28 September - 1 October 2021, Bergen, Norway, where this work was presented. This information has now been included in the reference. We also provide the DOI so that readers can easily find the published abstract.

We do present this work as one of several justifications for our study objectives and one of several possible interpretations of our findings. Please see response to comments 1 and 2 above, and 21 below.

4. They do provide additional citations and text in the discussion and some of that would really bolster the background section at the front of the paper.

Thank you for your suggestion. We have added some citations and text from our discussion to the introduction section (see response to comments 1 and 2).

5. I would have also liked to see more discussion of the setting (is it possible to provide a map of the surveyed areas?). It was not clear to me how proximate the survey data points are to each other (I'm unclear about what it means to be in a rural area in Bobo) or how proximate the data are to the temperature station (temp does spatially vary).

Unfortunately, it is not possible to provide a map of the surveyed areas as GPS data were not collected and residential addresses were not available for each participant. The weather station was located in the industrial district of the large urban centre (city of Bobo-Dioulasso). All participants lived within 40-50 km of the urban centre. We have included this information in the ‘Setting’ subsection of our methods (lines 145-148):

“Small settlements and villages, with a mainly agricultural focus, are located in rural areas surrounding the large urban centre. All rural participants in this study resided within 40-50 km of the city.”

Indeed, temperature does vary spatially, but 40-50 km is not an unreasonable area for exposure assignment, particularly given data limitations in low-income countries. Although absolute temperatures might differ by 1 or 2°C, daily fluctuations in temperature will be consistent over the study area. We have included this as a limitation in our discussion (lines 445-448):

“We used meteorological data recorded at one weather station in Bobo-Dioulasso, located in the urban centre. We could not assign exposures to women’s residential addresses, but daily variability in temperature exposures were likely to be consistent over the study area, even if absolute temperatures varied slightly.”

6. Relatedly, the authors briefly mention the source of the data - the PopDev program but they do not provide enough details on the data collection. Were the data collected in French (meaning only French speaking participants)?

Thank you for pointing this out. The data were collected in local languages (not in French), ensuring that all women identified as eligible could participate if willing. We have clarified this on lines 161-163:

“Interviews were conducted in local languages (predominantly Dioula) and homogeneity in translations was verified during interviewer training.”

7. How long did the surveys take?

Each interview lasted approximately 45-60 minutes. We have now included this information on lines 196-197:

“Interviews included additional questions to those described above in order to fulfil the aims of the PopDev study. Each interview lasted approximately 45-60 minutes.”

8. And, perhaps most importantly, how are these kinds of measurements (minutes of breastfeeding or time spent caring for children, for example) collected and validated?

Our tool was an adaption of a productivity cost tool tested by the IMMPACT project on maternal mortality in Burkina Faso, Indonesia and Ghana. <https://assets.publishing.service.gov.uk/media/57a08bd640f0b652dd000f14/prodcosts.pdf>. Our adaptation was based on expert knowledge of a senior Burkinabe economist in the team, who had carried out productivity studies, and the particularities of our research questions at the time of the original study. We pretested our survey instruments with a sample of women, and changed our tools accordingly. Some sections of our tools came from previous published studies we had conducted ourselves (see Filippi et al., 1997; Storeng et al., 2010: <https://doi.org/10.1016/j.socscimed.2010.03.056>), and others came from well know instruments such as DHS and K10. It was important to us that our instruments were easily understood and met closely our information needs for our research questions, therefore we prepared new questions when needed. We worked alongside qualitative scientists who also participated in our discussions in relation to the content of our quantitative tools. This is described in the participant and public involvement section.

Regarding how these data were collected, we have added a description (also in response to Reviewer 1’s comments) (lines 164-187):

“During each interview, participants were asked to recall their activities on the previous day (or two days previous when the day before interview was atypical) and how many minutes they had spent on each activity. The recall period was defined as, “between waking up yesterday morning and waking up this morning”. Participants described their activities and the interviewer categorised them using a pre-defined list, which was added to when needed. Breastfeeding, caring for children (bathing/dressing, feeding, playing/watching, tending to when unwell), income-generating work (e.g. agro-processing for trade, sale of products at market, small business, office activities), attending classes (e.g. literacy courses), and household chores (e.g. preparing meals, cleaning clothes, washing dishes, fetching water, fetching fuel) were included in this list.

To assist participants in time-use recall, women were initially asked to list all activities that they had engaged in during a specific time of day (e.g. between waking up and midday). Participants were then asked probing questions to assist them in estimating the duration of each activity. For example, some women said that they woke with the call of the muezzin, which enabled the interviewer to determine time of wake. Participants would then be asked if they began their first listed activity immediately, or if they did something else first. The interviewer asked when their first activity ended to which women responded, "to the first rays of sunshine", for example. Thus, the interviewer had adequate information to estimate time duration of the first activity. Once participants had finalised their time-use estimates for the specified period, the process was repeated for the next time of day, until the full 24-hour recall period was complete. Participants used a notebook to draft and revise time-use estimates before final responses were recorded in the questionnaire. Indications of time (particularly the path of the sun), combined with current events of life, served as benchmarks for time estimations."

9. Technically, it seems as though the question asks about how many minutes was a child "given the breast" - is this a culturally appropriate way of capturing breastfeeding duration?

This question was part of our adapted productivity tool, which was piloted with members of the community. Feedback on interview duration, meaningfulness and clarity of questions, and perceived gaps, was used to refine the wording of questions and to add/remove items. This information is included on lines 289-291.

10. Have these measures been used elsewhere? I was not able to find this information when I went through the website for the data either.

Please see the IMMPACT tool (link provided under comment 8), however questions on breastfeeding were not included. These questions were part of our adapted and piloted tool (see response to comment 8).

11. (Also, as a side note, the authors say that the list of tasks is "exhaustive" but when I looked at the survey it is not really exhaustive, but rather a relatively short list of questions about time use.)

We have removed the word "exhaustive", however the activities listed in the questionnaires are categories. We did not go into more detail on the questionnaires because of space constraints. Women explained their activities to the interviewers who categorized them. We have clarified this on lines 167-173 (new text in *italics*):

"Participants described their activities and the interviewer categorised them using a pre-defined list, which was added to when needed. Breastfeeding, caring for children (bathing/dressing, feeding, playing/watching, tending to when unwell), income-generating work (e.g. agro-processing for trade, sale of products at market, small business, office activities), attending classes (e.g. literacy courses), and household chores (e.g. preparing meals, cleaning clothes, washing dishes, fetching water, fetching fuel) were included in this list."

12. Because these measures are so key, it seems important that the authors spend more time addressing the details of the measures as well as address some of these issues in the limitations/conclusions sections.

We now include details of the breastfeeding and childcare duration measures (see response to comment 8 above). We also address the binary outcomes of exclusive breastfeeding and supplementary feeding (lines 189-195):

"At interview rounds two and three, women were asked if they were still breastfeeding their baby; if their baby had anything else to drink in the past 24 hours; and which (if any) fluids had their baby

been given to drink in the past 24 hours. These questions were used to construct binary study outcomes of exclusive breastfeeding (still breastfeeding and no fluids other than breast milk provided in past 24 hours) and supplementary feeding (any fluids other than breast milk provided in past 24 hours).”

We have included an ‘Outcomes’ subsection in the methods to define all study outcomes (lines 206-214):

“Four outcomes were assessed: (1) Breastfeeding duration: self-reported time spent breastfeeding on the day/night before interview (total minutes in 24 hours); (2) Exclusive breastfeeding: no liquids other than breast milk given in past 24 hours; (3) Supplementary feeding: any liquid other than breast milk given in past 24 hours; and (4) Childcare duration: self-reported time spent exclusively on childcare (including bathing/dressing, feeding, playing/watching, tending to when unwell) on the day/night before interview (total minutes in 24 hours). Breastfeeding duration, exclusive breastfeeding, and supplementary feeding outcomes referred specifically to the target (newborn) infant. Childcare duration did not refer specifically to the newborn infant.”

We discuss the limitations of our measures in the discussion section (lines 449-461):

“Measurements of breastfeeding and childcare duration relied on self-reported time-use estimations. We put several measures in place to assist participants, including the recent and short (24-hour) recall period with questions aimed at establishing a 24-hour timeline. However, it is possible that temperature affected participants’ ability to estimate time, despite the use of benchmarks (e.g. path of the sun). Time-use diaries and direct observation offer arguably more robust methods for future research, although each has limitations. Our measurement of exclusive breastfeeding was not optimal, but women were not asked directly if they breastfed their infant exclusively. Instead, this outcome was constructed from women’s recall of all fluids given to their child in the past 24 hours. Questions on infant feeding practices and childcare were embedded within an extensive interview schedule, further reducing the possibility of response bias. Finally, the outcome of childcare is complex and refers to time spent with children of all ages as this question was not specifically phrased to indicate the target (newborn) child.”

13. More citations and supporting text on what childcare means in this setting is vital - especially because this is sensitive to temperature and the authors do not really prepare us for why that might be with regard to what is actually being measured in this context.

Thank you for pointing this out. As noted above (comment 8), we now include the individual activities included under the outcome of childcare, i.e. “*bathing/dressing, feeding, playing/watching, tending to when unwell*”. We also include an ‘Outcomes’ subsection to clarify the meaning of childcare in our study (see response to comment 12).

14. The authors do address some of the issues around breastfeeding duration in the conclusion (e.g., the address the issue of fore vs. hindmilk). This might be helpful to also address in the beginning of the paper.

Thank you for this comment. We now address this issue in the Introduction (lines 85-88; see response to comment 1).

15. In terms of the temperature measurements, it was not clear to me why the authors used mean temp and not temp max or humidity (to get a "feels like").

We used daily mean temperature, rather than maximum temperature, because maximum temperature does not provide a good approximation of the temperature exposure over a 24-hour period. Maximum temperatures may be reached for an hour or less each day, whereas mean temperature accounts for

both maximum (daytime) and minimum (night-time) temperatures. We did, however, use apparent (or “feel like”) daily mean temperature within our sensitivity analysis. We realise that this was not clear in our original manuscript and, so, we have included this information in a new ‘Exposure’ subsection of our methods (lines 219-221):

“Apparent daily mean temperature (°C) was calculated from daily mean temperature, relative humidity and windspeed, using the R HeatStress package, [41] to test the robustness of our findings.”

We also state that apparent temperature was used in our sensitivity analysis on lines 272-274:

“Sensitivity analyses involved re-specifying models with [...] and (ii) apparent, rather than observed, daily mean temperature. [41]”

16. The authors also did not provide any justification for their temp measurement either.

Following on from the comment above, we did provide justification for our use of daily mean temperature, but we have moved this text to the ‘Exposure’ subsection to ensure it is clear to the reader (lines 216-218):

“The primary exposure was daily mean temperature (°C). Daily mean temperature correlated strongly with daily minimum ($r=0.8$, $p<0.001$) and maximum temperatures ($r=0.89$, $p<0.001$) and was considered the best approximation of overall exposure during the recall period.”

17. Considering alternative measures of temperature might be useful here.

Thank you for this comment. As described above, we used an alternative exposure (apparent daily mean temperature) in our sensitivity analysis. Use of “feels like” temperature did not change the estimated effect of temperature on breastfeeding duration, or exclusive breastfeeding of very young infants (< 3 months), but it did increase the statistical significance of the latter finding (a new analysis). We have included this information on lines 379-382:

“Redefining the exposure as apparent (“feels-like”) daily mean temperature did not change the estimated effect on exclusive breastfeeding of very young infants (< 3 months), but increased the statistical significance of this finding.”

18. (a small side note is that the authors do sometimes mention "heat stress" in the paper but really they are not measuring heat stress - rather they are measuring temperature and their results may have implications for how we think about heat stress, but heat stress is an individual-level biological process which is not measured here - please be careful on the use of the language around heat stress in the paper.

Thank you for this comment. We agree and have changed “heat stress” to “*high temperature*” (line 307).

19. The authors also mention that 32.5 c is "very hot" - it seems like that is relative - especially as compared to the north of Burkina).

Indeed, this is relative and has now been removed. However, a daily mean temperature of 32.5°C translates to a daily maximum temperature of ~40°C in this setting.

20. In terms of the results, I would have liked to see more discussion about how variable the temperature data actually was. It would be nice to know more about the distribution of the temperature data used in the analysis. Perhaps more discussion of Fig 3 discussing the difference (by infant age group) in duration for temps at the low and high end of the spectrum (with some indication of the count of observations). I generally think a discussion around the range of temps in the study and the associated outcomes would

help contextualize the findings. The discussion of the slope (e.g., change in duration associated with a 1 C increase) is less interesting than a discussion of the differences at the high and low end. I don't mean a discussion of the "hottest and coolest days" as in line 317 but more within a single wave of data collection (or among kids about the same age) how much variability did you really have in temp.

We include a figure showing the daily minimum and maximum temperatures throughout the study period in our supplementary material (figure S1). We also include the median and range of daily mean temperatures (recorded within each wave of data collection) in table 1, along with summary statistics for infant age stratified by interview round.

The range of temperatures was similar between interview rounds one (22.7 – 32.8°C) and two (22.9 – 33.7°C), with less variability at interview round three (23.3 – 30.3°C) – shown in table 1. We include the range in temperature for infants aged < 3 months and 3-6 months, reflecting our new stratified analysis of exclusive breastfeeding (lines 364-365):

“Variability in daily mean temperature was similar for both groups: < 3 months = 24.1–33.7°C; 3-6 months = 25.3–33.7°C, [...]”

21. I found it interesting that women with children under 6 months old did not really change their exclusive breastfeeding practices, even as temperatures increased and think the authors should discuss the underlying reasons they think may exist for this (as relates to their literature review). The authors mostly focused on the duration of the breastfeeding findings and a little around the supplementary feeding findings but some of the other findings are quite interesting as well.

Thank you for this comment. Following revisions based on Reviewer 1's comments, we now present findings on exclusive breastfeeding stratified by age (< 3 months, 3-6 months). We found suggestive evidence that the odds of exclusive breastfeeding very young infants (< 3 months) decreases as temperatures increase. Whereas, we found no evidence of association between temperature and exclusive breastfeeding of 3-6 month old infants. We now discuss these findings, as suggested, e.g. lines 408-418:

“The hot climate has been identified as a barrier to exclusive breastfeeding in the Democratic Republic of the Congo, [10] southern Zimbabwe, [13] Ghana, [12], and Ethiopia [50], and might (at least, partially) explain the very low rate of exclusive breastfeeding found herein. In the cooler subtropical climate of Bihar, India, [30] odds of exclusive breastfeeding was significantly lower in summer than in winter or transitional seasons. Perceived thirst was proposed as an underlying cause for the higher rates of supplementary feeding in warmer months. [30] However, in contrast to our findings, the impact of season was greater for infants aged 3-6 months than for infants under 3 months. [30] The warmer climate of Bobo-Dioulasso, cultural values and beliefs around breastfeeding, [54] and the comparatively low rate of exclusive breastfeeding in our study (8% vs. 70% of infants aged 3-6 months) likely explain this discrepancy in findings”

22. For the limitations section - is there any possibility of addressing anything like co-nursing or other adults in the house who might help with care/nursing? I'm not sure if this is relevant in this setting. I also wonder about the use of formula - is it possible that individuals are using formula? That seems like quite a difference in health impacts as opposed to something like water. In other words - there are good alternatives to breastfeeding that women may rely on and not all supplements should be considered as equal. Again, I'm not sure if this is relevant in this setting but it seems that women should always have access to high quality alternatives to breastfeeding as part of ensuring women's autonomy and equality.

Thank you for this considered comment. Women were not asked about co-nursing. We did ask women if they had resumed all their normal activities since giving birth and if they had to ask for help. However, this question was directed more towards work activities (professional and domestic) than breastfeeding or childcare. We included living arrangements in the childcare model, and found that women who were not in a relationship spent significantly less time on childcare than women who lived with their partner full-time (table S3 in supplementary material), presumably because they had less help. However, living arrangements was not a significant covariate in the breastfeeding models.

Focussing on women with infants under 6 months, the provision of powdered formula or other non-breast milks was quite uncommon (30 of 715 women), compared to water (569) or herbal teas (357 women). Also, of those 30 women who used alternative milks, the vast majority (28/30) also reported providing water or herbal teas.

Reviewer #1

Competing interests of Reviewer: NA

Reviewer #2

Competing interests of Reviewer: none

VERSION 2 – REVIEW

REVIEWER	Cliffer, Ilana Harvard University T H Chan School of Public Health, Global Health and Population
REVIEW RETURNED	16-Aug-2022
GENERAL COMMENTS	Thank you to the authors for their thorough responses to all of my previous comments and questions, including their re-analysis of several aspects of the manuscript. I have no further comments!